# ECFCON: Emotion Consequence Forecasting in Conversations

## ABSTRACT

Conversation is a common form of human communication that includes extensive emotional interaction. Traditional approaches focused on studying emotions and their underlying causes in conversations. They try to address two issues: what emotions are present in the dialogue and what causes these emotions. However, these works often overlook the bidirectional nature of emotional interaction in dialogue: utterances can evoke emotions (*cause*), and emotions can also lead to certain utterances (*consequence*). Therefore, we propose a new issue: what consequences arise from these emotions? This leads to the introduction of a new task called Emotion Consequence Forecasting in CONversations (ECFCON). In this work, we first propose a corresponding dialogue-level dataset. Specifically, we select 2,780 video dialogues for annotation, totaling 39,950 utterances. Out of these, 12,391 utterances contain emotions, and 8,810 of these have discernible consequences. Then, we benchmark this task by conducting experiments from the perspectives of traditional methods, generalized LLMs prompting methods, and clue-driven hybrid methods. Both our dataset and benchmark codes are openly accessible to the public.

## CCS CONCEPTS

• **Information systems** → **Sentiment analysis**; • **Computing methodologies** → **Natural language processing**.

## KEYWORDS

Emotion Consequence, Conversations, Multimodal, Dataset

**ACM Reference Format:**
Anonymous Author(s). 2024. ECFCON: Emotion Consequence Forecasting in Conversations. In *Proceedings of the 32th ACM International Conference on Multimedia (MM '24)*. ACM, New York, NY, USA, 9 pages. https://doi.org/XXXXXXX.XXXXXXX

## 1 INTRODUCTION

Emotions play a pivotal role in human communication, influencing not only personal interactions but also the effectiveness of human-computer interfaces. Understanding and predicting emotional dynamics is therefore crucial for developing human-like AI systems. Although significant progress has been made in emotion analysis within the field of multimodal natural language processing, forecasting the consequences of emotions in conversations remains a considerable challenge. To bridge this gap, we introduce

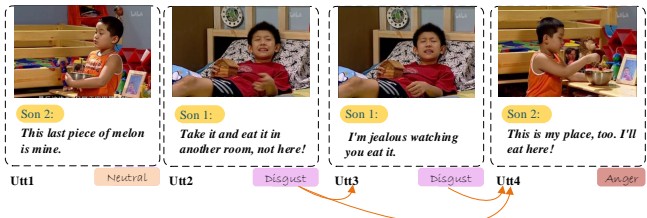

**Figure 1: A sample of the annotated conversation in ECFCON: Two kids arguing about eating watermelon.**

a novel task called Emotion Consequence Forecasting in CONversations (ECFCON). This task focuses on examining how emotions evolve and impact subsequent interactions, aiming to enhance the emotional intelligence of AI systems.

Dialogue is a common form of human interaction, often characterized by shifting emotions. This type of emotional interaction, which is predominantly bidirectional, is rarely found in non-dialogue scenarios. An important yet overlooked *phenomenon* is that some utterances can evoke emotions, while emotions can also inspire subsequent utterances. Previous researches in conversations have mainly focused on emotion recognition [16, 25, 29, 35, 37, 38] and investigating what causes emotions [5, 6, 15, 30, 31]. However, the subsequent impact of emotions in dialogues has not been adequately addressed.

Figure 1 presents an example of such emotional consequences. In the scenario, two children are arguing over a watermelon that belongs to Son 2. Son 1 is *disgust*ed and expresses a desire not to have the watermelon in his sight. Consequently, after expressing his *disgust* and asking Son 2 to leave, Son 1 reveals his underlying jealousy. This emotion prompts a strong reaction from Son 2, who insists on staying in the room to eat. In this sequence, the utterance expressing *disgust* (Utt2) leads to the consequences seen in Utt3 and Utt4, and the continued expression of *disgust* in Utt3 leads directly to Utt4.

Our dataset is annotated from dialogical videos as interactions in videos are more distinct than those in pure text. Moreover, multimodal information such as body language, facial expression changes, and tone of voice variations facilitates easier identification of the consequences of emotions.

To the best of our knowledge, this task is mostly the first attempt in this research area. Specifically, we summarize our contributions as follows:

1. We introduce a new task, named **Emotion Consequence Forecasting in CONversation**, explaining the rationale and starting point of the task. In particular, we define the types of emotional consequences and clarify the differences between the emotional consequences and causes.(Section 3)

2. We detail the dataset annotated for this task, ECFCON, which includes 2,780 video dialogues, 39,950 utterances, 12,391 of which are emotional utterances, and 8,810 of these emotional utterances

have consequences. To the best of our knowledge, this is the first dataset for this task. (Section 4)

3. We experiment with different approaches on our dataset from the perspectives of traditional methods, generalized LLMs prompting methods, and clue-driven hybrid methods. Our experiments show that clue-driven hybrid methods outperform other baselines. (Section 5)

## 2 RELATED WORK

In this section, we review the related works of emotion analysis in conversation from two perspectives: datasets and approaches.

### 2.1 Datasets

Emotion analysis in conversation has been a focus of recent research. Several datasets have been developed for emotion recognition, including MELD [23], IEMOCAP [3], SEMAINE [22], DailyDialog [17], EmoContext [4], MELSD [10]. These datasets have primarily focused on identifying emotions within conversations. With the introduction of RECCON [24] for recognizing emotion causes in conversations, research has begun to explore the cause of emotions. Following this work, Wang et al. [31] proposed a multimodal emotion cause dataset, ECF. Our work starts from the perspective of the interaction in conversation and extends this line of research by introducing ECFCON, a dataset designed to forecast the consequences of emotions in conversations, thereby addressing the deficiency in the current research landscape.

### 2.2 Approaches

The approaches in emotion analysis in conversation have primarily focused on emotion recognition and cause extraction. For emotion recognition, a variety of approaches have been proposed in recent years, such as Ishiwatari et al. [14], Li et al. [16], Lian et al. [18], Lu et al. [21], Qin et al. [25], Shen et al. [28], Shi and Huang [29], Wang et al. [34], Zhang et al. [35], Zhang and Li [37], Zhao et al. [38, 39]. These approaches aim to identify the indicative emotions of utterances. For cause extraction, research has been focused on determining what causes emotions in conversations, with notable works Chen et al. [5, 6], Gao et al. [11], Jeong and Bak [15], Singh et al. [30], Wang et al. [31]. However, these approaches have not addressed the issue of forecasting the consequences of emotions in conversations, which is the primary focus of our work.

## 3 TASK DEFINITION

As this task is newly introduced, we first clarify its definition.

**Emotion** is a psychological state associated with mood, behavior, and responses, and it represents the state in the process of human interaction [9]. In computer science, Ekman's six universal emotions—*anger*, *disgust*, *fear*, *happiness*, *sadness*, and *surprise*—are often commonly used as the basis for emotion recognition [1]. Additionally, the *neutral* label is often included. In conversation, most existing works have annotated the indicative emotions of utterances using this same framework [3, 17, 23].

**Consequence** refers to a result or effect of an action or condition. In conversation, the emotion consequence denotes the subsequent utterances that are directly inspired by the indicative emotion of a previous utterance. These consequences can manifest as verbal

responses such as affirmations or confrontations, physical actions including evasion, approaching, trembling, jumping, etc., and also changes in tone of voice.

**Consequence Types** refer to the categories of consequences that are inspired by the indicative emotion of an utterance. We define the types of consequences as follows:

*objective*: These consequences are objective and primarily involve physical actions, such as evasion, approaching, trembling, jumping, as well as verbal responses, including plans, and vocal responses such as tone changes.

*subjective*: These consequences are subjective and primarily consist of responses, confrontations, affirmations, standpoints, etc.

*What is the difference between the emotion cause and consequence?* The emotion cause is the reason why the emotion is generated, while the emotion consequence is the result of the emotion. Typically, the causes are found mostly in previous utterances, and the consequences are in subsequent utterances. Furthermore, the causes often consist of a few scattered utterances that trigger the emotion, whereas the consequences comprise a series of consecutive utterances that are directly related to the emotion.

*Why the emotion consequences are not discrete but consecutive utterances?* As illustrated in Figure 1, the emotion consequences of Utt2 are Utt3 and Utt4, and the emotion consequence of Utt3 is Utt4. The emotion consequences are closely related to the emotional utterance. If there is more than one consequence, they start from the next utterance following the emotional utterance and continue consecutively to the end. This pattern occurs because the cause is typically clear, consisting mainly of utterances that trigger emotions, depending specifically on that particular utterance. Although some utterances are close to the emotional utterance, they are not the reason for the emotion. In contrast, the consequences are diverse in type. Generally, subsequent utterances all respond to the emotional utterance. Once the emotional responses subside, the impact of the emotion ends, and the conversation shifts to a new topic.

Then, we define three sub-tasks of ECFCON. Given a video conversation $D=\{U_1, U_2, \ldots, U_n\}$, where $U_i$ is the $i$-th utterance in the conversation, and each utterance contains three modalities: **t**ext, **a**udio, and **v**ideo, defined as $U_i = \{U_i^t, U_i^a, U_i^v\}$. The three sub-tasks are as follows:

**Task1**: *Consequence Forecasting (CF)*, inferring the consequences of the emotional utterance $U_i$ in $D$.

$$P_{\{i+1:n\}}^c = \{y_{\{i+1:n\}}^c | U_i, e_i, D\} \quad (1)$$

where $P_{\{i+1:n\}}^c$ refers to the probability distribution that the utterances from $i+1$ to $n$ are the consequences of the emotional utterance $U_i$. $e_i$ is the emotion of $U_i$ and $y^c$ is consequence category.

**Task2**: *Emotion Consequence Pair Forecasting (ECPF)*, inferring the emotion-consequence pairs in $D$. The process first identifies the emotional utterances and then forecasts the consequences of these emotional utterances.

$$P_i^e = \{y^e | U_i, D\} \quad (2)$$
$$P_{\{i+1:n\}}^c = \{y_{\{i+1:n\}}^c | U_i, D\} \quad (3)$$

where $P_i^e$ refers to the probability distribution and $y^e$ is the emotion category. if $U_i$ is an emotional utterance, then $P_{\{i+1:n\}}^c$ is the

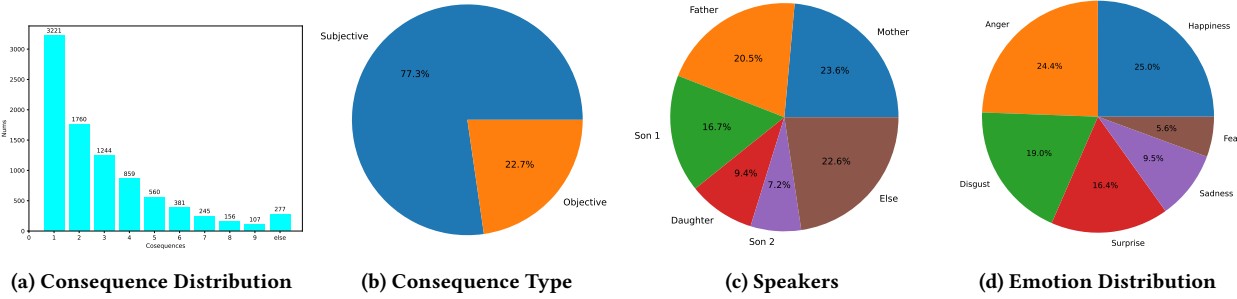

| (a) Consequence Distribution | (b) Consequence Type | (c) Speakers | (d) Emotion Distribution |

**Figure 2: Data Analysis.**

probability distribution that the utterances from $i + 1$ to $n$ are the consequences of the emotional utterance $U_i$.

**Task3**: *Emotion Consequence Pair Forecasting with Categories (ECPF-C)*. Distinguishing from ECPF, ECPF-C needs to recognize the fine-grained emotion category of the utterances. Specifically, in equation 2, the $y^e$ contains seven types of emotion.

## 4 BUILDING THE ECFCON DATASET

### 4.1 Video Dialogue Sources

We collected video dialogues from a Chinese situation situation comedy, named *Home with Kids*, which are rich in emotional interactions. This series is similar to the American sitcom *Friends*, which has been used as a source for datasets such as MELD [23] and ECF [31]. We selected 100 episodes from *Home with Kids*, each approximately 25 minutes in length. From these episodes, we identify dialogues that contain emotional interactions and ultimately collect 2,780 video dialogues. Some dialogues are excluded from the source due to unsuitability for annotation, such as those featuring only one speaker or lacking any emotional interactions.

### 4.2 Data Annotation

*Annotators*: we recruited 10 annotators, all of whom are graduate and undergraduate students majoring in computer science. They are trained to understand the definition of the task and the annotation guidelines.

*Annotation platform*: we develop a web-based annotation platform specifically for this task. This platform greatly improves the efficiency and enhances the quality of the annotation. The annotation platform will be released in the future.

*Annotation Rewards*: We provide the annotators with reasonable and generous wages, allowing them to focus on annotating data, thereby ensuring high-quality annotations.

### 4.3 Quality Assessment

To evaluate the quality of the annotated dataset, we applied Cohen's Kappa [7], a widely accepted metric for measuring inter-annotator agreement. Each utterance was initially annotated by two annotators with expertise in the relevant subject area. Any discrepancies in annotations are identified and resolved by a third expert, who makes the final judgment. As indicated in Table 1, the Kappa score for emotion is 0.8553, demonstrating a high level of agreement between the annotators. Similarly, the Kappa score for consequence is 0.7699, indicating substantial agreement.

**Table 1: Inter-annotator agreement.**

| Items | Kappa |
|---|---|
| Emotion | 0.8553 |
| Consequence | 0.7699 |

### 4.4 Dataset Statistics

As shown in Table 2, the dataset comprises 2,780 video dialogues with an average duration of 34.9 seconds. It includes 39,950 utterances, of which 12,391 have indicative emotions. Among these, 8,810 emotional utterances have corresponding consequences.

Figure 2 displays the distribution of the dataset. The consequence distribution is illustrated in (a). It can be observed that the number of consequences associated with a single emotion is typically concentrated in the range of 1-5. As the number of consequences increases, the frequency of emotion-consequence pairs decreases. The distribution of consequence types is presented in (b), where most of the consequences are subjective (77.3%), and objective consequences comprise 22.7%. (c) shows the speaker distribution, revealing that the majority of the speakers belong to one family, with only 22.6% from speakers outside the family. The emotion distribution is depicted in (d), which is relatively balanced, with *happiness* (25.0%), *anger* (24.4%), and *disgust* (19.0%) being the most prevalent emotions.

**Table 2: Basic statistic of our ECFCON dataset.**

| Items | Number |
|---|---|
| Avg video duration | 34.9s |
| Conversations(videos) | 2780 |
| Utterances | 39950 |
| Emotion(utterances) | 12391 |
| Emotion(utterances) with consequences | 8810 |

## 5 METHODOLOGY

Due to the lack of previous work in this area, we propose a series of baseline approaches from three perspectives: traditional methods, generalized LLMs prompting, and clue-driven hybrid methods. By constructing these baselines, we aim to provide a comprehensive benchmark for the ECFCON dataset, facilitating a better understanding of this task.

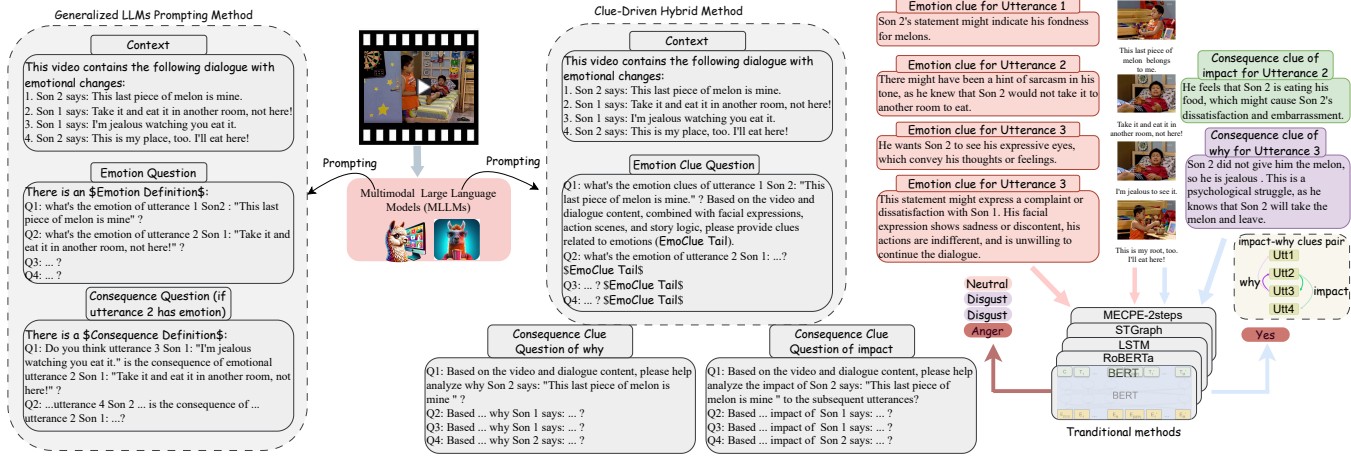

**Figure 3: The framework of our proposed methods.**

## 5.1 Traditional Methods

***Dialogue feature extraction.*** We extract the dialogue features from the text, audio, and video modalities and then concatenate them to form the multimodal representation of each utterance $u_i = \{u_i^t, u_i^a, u_i^v\}$.

• *Text*: We initialize the tokens and then feed them into the LSTM to obtain the textual representation of each utterance $u_i^t$. In addition to it, we also attempt to use the pre-trained BERT [8], RoBERTa [20] to extract the text features.

• *Audio*: We extract the audio features using the HuBERT [12], which is a pre-trained model for audio feature extraction.

• *Video*: We apply the visual encoder of the pre-trained CLIP [26] to extract the visual features, which may migrate the gap between the text and video.

***Emotion Consequence Forecasting.*** Due to the lack of contextual relationships between the utterances and the integration of multimodal information, we feed the independent utterance representations $u_i$ into the utterance-level encoder, which can be LSTM, Transformer, etc. Then, the hidden states of the encoder $h_i$ are fed into modules for emotion recognition and consequence forecasting.

For *Task1: Consequence Forecasting (CF)*, we obtain the probability distribution of the consequences of utterance $U_i$ (assuming $U_i$ is an emotional utterance) as follows:

$$
\begin{aligned}
E_i &= \text{Embedding}(e_i) & (4)\\
h_i^e &= W^e[h_i; E_i] + b^e & (5)\\
h^c &= W^c h + b^c & (6)\\
P_{\{i+1:n\}}^c &= \text{MLP}^c([h_{\{i+1:n\}}^c; h_i^e]) & (7)
\end{aligned}
$$

where $E_i$ is the embedding of emotion $e_i$, $h_i^e$ is the representation of emotional utterance, $h^c$ is the representation of consequence utterance. MLP is a multi-layer perceptron network. $P_{\{i+1:n\}}^c$ is the probability distribution of the consequences from $i + 1$ to $n$.

For *Task2: Emotion Consequence Pair Forecasting (ECPF)*, we first recognize the emotion of the utterance and then forecast the consequences of the emotional utterance. Here, we assume that utterance

$U_i$ is predicted as an emotional utterance, and then we forecast the consequences of $U_i$ as follows:

$$
\begin{aligned}
h_i^e &= W^e h_i + b^e & (8)\\
h^c &= W^c h + b^c & (9)\\
P_i^e &= \text{MLP}^e(h_i^e) & (10)\\
P_{\{i+1:n\}}^c &= \text{MLP}^c([h_{\{i+1:n\}}^c; h_i^e]) & (11)
\end{aligned}
$$

where $P_i^e$ is the probability distribution of the emotion of $U_i$. $P_{\{i+1:n\}}^c$ is the probability distribution of the consequences of $U_i$ from $i + 1$ to $n$.

For *Task3: Emotion Consequence Pair Forecasting with Categories (ECPF-C)*, the process is similar to ECPF as equation (8-11), but we need to predict the fine-grained emotion category.

***Loss Calculation.*** We use the cross-entropy loss to calculate the loss of the emotion and consequence forecasting as follows:

$$
\mathcal{L}^e = -\sum_{i=1}^{n} y_i^e \log P_i^e \tag{12}
$$

$$
\mathcal{L}^c = -\sum_{i=1}^{n} \sum_{j=i+1}^{n} \hat{e}_i y_j^c \log P_j^c \tag{13}
$$

$$
\mathcal{L} = \alpha \mathcal{L}^e + \beta \mathcal{L}^c \tag{14}
$$

where $\mathcal{L}^e$ is the emotion loss, $\mathcal{L}^c$ is the consequence loss, $y_i^e$ is the ground truth emotion, and $y_j^c$ is the ground truth consequence of $U_i$. $\hat{e}_i$ is the indicator function, which is 1 if $U_i$ is an emotional utterance, and 0 otherwise. $\alpha$ and $\beta$ are hyperparameters to balance the emotion and consequence losses.

## 5.2 Generalized Prompting Methods

Since the Large Language Models (LLMs) have shown strong performance in various NLP tasks, we also attempt to solve our tasks in a generative manner. As our dataset is dialogue-based videos, we try to use the Multimodal video-based LLMs (MLLMs), such as Video-LLaVA [19], Video-LLaMA[36]. We first input the video into MLLMs and then build the prompts for our tasks, as shown in the left part of Figure 3. The prompts are constructed as follows:

*Context* contains a prefix and the sequences, speaker names, and the text of the utterances of the dialogue.

*Emotion Question* is the prompt for the emotion recognition. It is constructed to obtain the answer for the emotion category.

*Consequence Question* is the prompt for the consequence forecasting. It is constructed to obtain the answer of whether the utterance is the consequence of the emotional utterance.

This prompting uses a generalized template, which has been proven to be effective in various tasks, [2, 27, 32, 40].

$$A_i^e = argmaxp(\mathcal{A}_e|u_i, D) \tag{15}$$
$$A_{ij}^c = argmaxp(\mathcal{A}_c|u_i, e_i, u_j, D) \tag{16}$$

where $A$ represents the final answer, generated from all potential answers $\mathcal{A}$. $u_i$ is the potential emotional utterance, and $u_j$ is the potential consequence utterance of $u_i$. $e_i$ is the emotion of $u_i$.

## 5.3 Clue-Driven hybrid Methods

Due to the poor performance of the generalized prompting with MLLMs, and the high computational cost of fine-tuning the MLLMs, we attempt to combine traditional methods with MLLMs. The MLLMs contain rich knowledge and have strong reasoning abilities, which may help to improve the performance of the traditional methods. To this end, we propose a clue-driven hybrid method, which resorts the MLLMs to provide the clues for the traditional methods, as shown in the right part of Figure 3.

*Emotion clue generation*: We first input the dialogue video and the complete content of the dialogue into the MLLMs and then attempt to extract the emotion clues from the generated answers. Through the setting of the prompting template, we resort to the MLLMs to generate emotional clues from the perspectives of facial expressions, action scenes, story logic, etc. These clues can be considered as multimodal clues, which compared to traditional feature extraction methods [12, 26], are simpler and more effective, and can greatly alleviate the burden of modality heterogeneity.

*Consequence clue generation*: We start from two directions: forward(*why*) and backward (*impact*) to find suitable clues

- *Impact* means a marked impression or effect on someone's feelings or thoughts. Since LLMs have relatively rich knowledge and strong reasoning ability, we hope to infer the potential impact of the emotional utterance on the subsequent utterances.
- *Why* means the reason or explanation for something. Here, we hope to infer the potential reason for making the subsequent utterances.

These impact-why clues pair can form a forward and backward loop of clues, which can provide the traditional methods with more effective clues. This loop is shown in the yellow part of Figure 3. *Impact* clues indicate how emotional utterances affect subsequent utterances, while *why* clues indicate the reason in previous utterances that lead to the current response. This forward and backward loop aligns well with the nature of the conversation, which is a continuous process of interaction. Besides, this also can maximally mine the conversational common sense knowledge in MLLMs and understand the logic between the proceedings and following parts of the conversation.

*Clue-driven forecasting*: Subsequently, we input the clues into the traditional methods to forecast the emotion and consequences. We first encode them as clue representations with a similar encoder of text and then concatenate them with the utterance representations as $u_i = \{u_i^t, u_i^a, u_i^v, c_i^e, c_i^{wh}, c_i^{im}\}$.

Then, the emotion recognition can be formulated as:

$$h_i^e = W^e[h_i; c_i^e] + b^e \tag{17}$$
$$P_i^e = \text{MLP}^e(h_i^e) \tag{18}$$

Here, the emotional clues have been integrated into the emotion recognition process. And the consequence forecasting of emotional utterance $u_i$ can be formulated as:

$$\hat{h}_i^e = W^e[h_i; c_i^{im}] + b^e \tag{19}$$
$$\hat{h}^c = W^c[h; c^{wh}] + b^c \tag{20}$$
$$P_{\{i+1:n\}}^c = \text{MLP}^c([\hat{h}_{\{i+1:n\}}^c; \hat{h}_i^e]) \tag{21}$$

Here, we can see that the *impact* clue is integrated into the emotional utterance, while the *why* clue is integrated into the target consequence utterance. This interweaving of clues may help find the connection between emotion and consequences.

## 6 EXPERIMENTATION

### 6.1 Experimental Setup

**Implementation Details.** The experiments are conducted on a server with 8 NVIDIA V100 GPUs and 2 GeForce RTX 4090 GPUs and implemented in PyTorch. We divide the dataset into training, validation, and testing sets at a ratio of 8:1:1 at the conversation level. Our traditional and hybrid models are trained on Adam optimizer, with a batch size of 64. The learning rate is 5e-5 for trained modules and 5e-8 for pretrained modules, such as BERT [8], RoBERTa [20]. We train each model and each task for 50 epochs and monitor its performance on the validation set. After training, we select the model with the best performance on the validation set and evaluate it on the test set.

From the Huggingface[1], we download *bert-base-chinese*, *chinese-hubert-base*, *clip-vit-large-patch14*, *roberta-base-finetuned-chinanews-chinese*, *Video-LLaVA-7B* and *LanguageBind Video merge*. These pretrained models are used in our experiments.

For **prompting**, we select the Video-LLaVA [19] as the MLLMs, which is a multimodal video-based LLMs and has shown strong performance in various tasks. The model supports an input length of 2k, which is sufficient to accommodate the entire conversation. Besides, it supports Chinese language input, which is suitable for our dataset. When few-shot learning, we use the Lora method [13] to fine-tune the MLLMs with 4 bits setting. Due to the enormous computational cost of fine-tuning the MLLMs, which is unaffordable for us, we only perform fine-tuning for fewer than 100 samples.

**Evaluation Metrics.** We evaluate the performance of the models using the precision (P), recall (R), and F1-score (F1) metrics.

### 6.2 Baselines

- MECPE-2steps [31] introduces a new task named Multimodal Emotion-Cause Pair Extraction in Conversations, along with a multimodal conversational emotion cause dataset. Although this task is

---

[1] https://github.com/huggingface

**Table 3: The performances of different baselines on the ECFCON dataset.**

| Method | CF | | | ECPF | | | ECPF-C | | |
|---|---|---|---|---|---|---|---|---|---|
| | P | R | F1 | P | R | F1 | P | R | F1 |
| MECPE-2steps (Text) | 0.6601 | 0.5539 | 0.6024 | 0.4748 | 0.1658 | 0.2458 | 0.2574 | 0.0584 | 0.0952 |
| + **Audio** | 0.5823 | 0.6516 | 0.6150 | 0.4446 | 0.1725 | 0.2486 | 0.2550 | 0.0662 | 0.1052 |
| + **Video** | 0.6340 | 0.5853 | 0.6087 | 0.3081 | 0.2425 | 0.2714 | 0.1966 | 0.0872 | 0.1208 |
| + **Audio** + **Video** | 0.5818 | 0.6493 | 0.6137 | 0.3466 | 0.2452 | 0.2872 | 0.1877 | 0.1257 | 0.1506 |
| + **Audio** + **Video** + clues | 0.6142 | 0.6246 | **0.6194** | 0.3230 | 0.3398 | **0.3312** | 0.1904 | 0.1475 | **0.1662** |
| ECFCON-STGraph (Text) | 0.6369 | 0.5554 | 0.5934 | 0.3263 | 0.2302 | 0.2699 | 0.1141 | 0.0685 | 0.0856 |
| + **Audio** | 0.5626 | 0.6299 | 0.5944 | 0.3121 | 0.2829 | **0.2968** | 0.2050 | 0.0700 | **0.1043** |
| + **Video** | 0.6023 | 0.5097 | 0.5522 | 0.2821 | 0.2919 | 0.2869 | 0.0873 | 0.0891 | 0.0882 |
| + **Audio** + **Video** | 0.6042 | 0.5771 | 0.5903 | 0.3005 | 0.2856 | 0.2928 | 0.0689 | 0.1100 | 0.0848 |
| + **Audio** + **Video** + clues | 0.6105 | 0.5913 | **0.6008** | 0.2498 | 0.2115 | 0.2290 | 0.2697 | 0.0177 | 0.0332 |
| ECFCON-LSTM (Text) | 0.5196 | 0.7305 | 0.6072 | 0.3391 | 0.1901 | 0.2436 | 0.1579 | 0.0749 | 0.1015 |
| + **Audio** | 0.5776 | 0.6549 | 0.6138 | 0.3727 | 0.2257 | 0.2811 | 0.0881 | 0.1306 | 0.1052 |
| + **Video** | 0.6238 | 0.6032 | 0.6134 | 0.3323 | 0.2706 | 0.2983 | 0.1689 | 0.1254 | 0.1439 |
| + **Audio** + **Video** | 0.5751 | 0.6695 | 0.6187 | 0.3325 | 0.3084 | **0.3200** | 0.1725 | 0.1153 | 0.1382 |
| + **Audio** + **Video** + clues | 0.5560 | 0.7148 | **0.6255** | 0.2725 | 0.3866 | 0.3197 | 0.1553 | 0.1519 | **0.1536** |
| ECFCON-BERT (Text) | 0.6210 | 0.5973 | 0.6089 | 0.3871 | 0.2092 | 0.2716 | 0.2386 | 0.0902 | 0.1309 |
| + **Audio** | 0.5564 | 0.7129 | 0.6250 | 0.3930 | 0.2672 | 0.3181 | 0.2411 | 0.1018 | 0.1432 |
| + **Video** | 0.5884 | 0.6437 | 0.6148 | 0.2879 | 0.2710 | 0.2792 | 0.1836 | 0.1235 | 0.1477 |
| + **Audio** + **Video** | 0.5356 | 0.7644 | 0.6299 | 0.3418 | 0.3013 | 0.3203 | 0.2149 | 0.1216 | 0.1554 |
| + **Audio** + **Video** + clues | 0.5980 | 0.6755 | **0.6344** | 0.3505 | 0.3484 | **0.3495** | 0.1875 | 0.1737 | **0.1803** |
| ECFCON-RoBERTa (Text) | 0.5640 | 0.6302 | 0.5953 | 0.4072 | 0.2047 | 0.2725 | 0.2428 | 0.0827 | 0.1234 |
| + **Audio** | 0.5635 | 0.6879 | 0.6195 | 0.3335 | 0.3237 | 0.3286 | 0.2516 | 0.1025 | 0.1457 |
| + **Video** | 0.5625 | 0.6688 | 0.6110 | 0.3512 | 0.2867 | 0.3157 | 0.2027 | 0.1033 | 0.1368 |
| + **Audio** + **Video** | 0.6096 | 0.6329 | 0.6210 | 0.3082 | 0.3900 | **0.3443** | 0.1742 | 0.1392 | 0.1548 |
| + **Audio** + **Video** + clues | 0.5511 | 0.7320 | **0.6288** | 0.3208 | 0.3447 | 0.3323 | 0.1932 | 0.1467 | **0.1668** |
| ECFCON-MLLMs (zero-shot) | 0.4006 | 0.5011 | 0.4453 | 0.2239 | 0.1841 | 0.2021 | 0.0611 | 0.1062 | 0.0776 |

different from ours, it is the most relevant work to ours. Therefore, we modified this approach to adapt it to our task. This approach is also based on BERT as the foundation.

• STGraph [33] introduces a Fully-Connected Spatial-Temporal Graph Network for Multivariate Time Series Data, which may be suitable for our task. We chose this method to explore the potential of graph-based methods in our task. This is a relatively advanced graph method, which can capture the temporal relationship between the utterances and the connections between modalities in spatial terms.

• ECFCON-LSTM, ECFCON-BERT, and ECFCON-RoBERTa are the traditional methods with different text feature extraction methods. We use a relatively general fusion method for these baselines to ensure the fairness of the comparison.

• ECFCON-MLLMs mainly uses Video-LLaVA [19] as the base and employs the template for emotion consequence forecasting. Few-shot learning is also based on this model.

## 6.3 Experimental Results

In this section, we present the experimental results of the baselines on the ECFCON dataset in Table 3.

*6.3.1* ***Baselines Comparison.*** Traditional and hybrid methods have demonstrated relatively good performance on the ECFCON

dataset. Overall, the BERT-based method achieved the best performance across the three tasks, with F1 scores of 0.6344, 0.3495, and 0.1803 respectively.

• Firstly, MECPE-2steps [31] does not perform well because it is primarily designed for identifying the causes of emotions, and can directly predict the causes without needing to first recognize the emotions. Despite being also BERT-based and modified for our task, it struggles with predicting consequences without emotion recognition.

• Secondly, ECFCON-STGraph does not perform well because it focuses solely on the fusion of modalities and temporal relationships, overlooking the critical aspect of the emotion-consequence relationship. Additionally, its use of convolution layers for graph information extraction may not be optimal for our task.

• Thirdly, ECFCON-LSTM performs not well due to its weaker ability to construct the text features compared to BERT and RoBERTa.

• Finally, although RoBERTa uses more diverse pre-training data and more advanced training strategies compared to BERT, it has not shown significant improvement in our task. we deduce that this may be due to two reasons: 1) the quality of the Chinese RoBERTa is inferior and its pre-training process was less comprehensive compared to the English RoBERTa 2) The source of the pre-training data is inconsistent with that of BERT, and the bias in data may lead to a decline in performance.

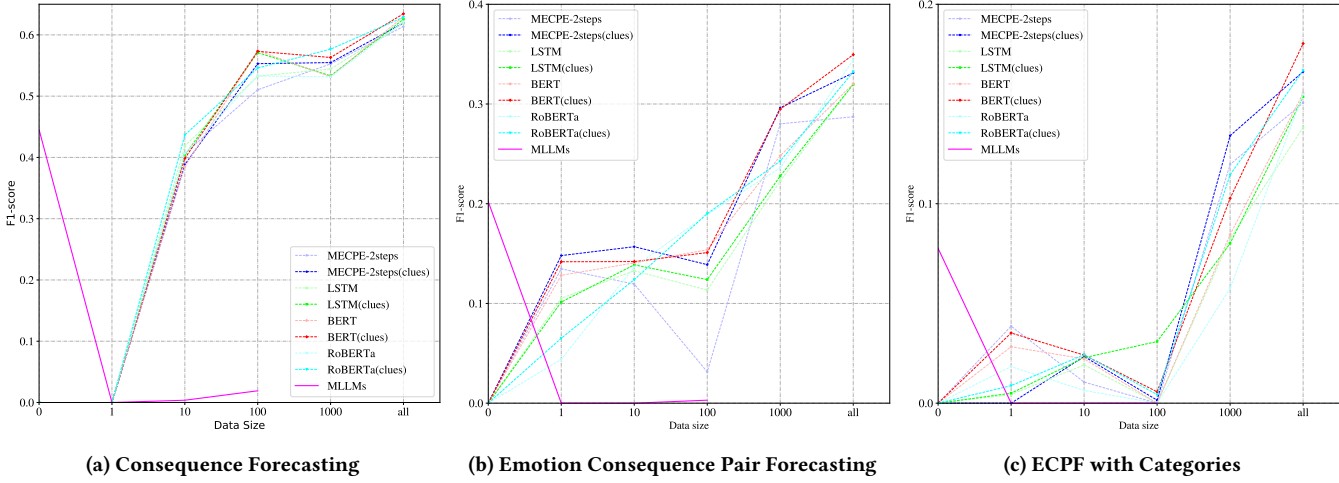

(a) Consequence Forecasting   (b) Emotion Consequence Pair Forecasting   (c) ECPF with Categories

**Figure 4: Few-shot learning**

*6.3.2* ***Modality Effectiveness.*** To explore the effectiveness of different modalities in our ECFCON dataset, we compare the results of various methods before and after adding the acoustic and visual information.

We can observe that most methods demonstrate significant improvements in F1 scores after adding the acoustic and visual information. This is particularly evident with ECFCON-BERT and ECFCON-RoBERTa, where the addition of audio, video, or both, leads to increased F1 scores across all three tasks. Methods that utilize text, audio, and video combined outperform those that include only text and audio, text and video, or solely text. These findings suggest that the multimodal information is largely complementary and enhances performance in our task.

Meanwhile, for MECPE-2steps, ECFCON-STGraph, and ECFCON-LSTM, some cases have shown a decline in performance after adding the acoustic and visual information. This can be attributed to the following reasons:

• Firstly, the performance of ECFCON-STGraph is highly fragmented and chaotic. This issue arises because the graph convolution of STGraph is not well-suited for modality fusion, and it struggles with managing long-distance temporal relationships. Additionally, the model has difficulty processing the forward and backward relationships between emotions and consequences.

• Secondly, only in the CF task of MECPE-2steps and in the ECPE-C task of ECFCON-LSTM do the three-modal methods fail to show the best performance. In other instances for these two methods, the F1 scores are generally normal, which also proves the effectiveness of multimodal information. One possible explanation is that in the multi-party dialogues within ECFCON, visual and acoustic features can become entangled, leading to a significant influx of noise. Another reason is that these are just baseline systems. The construction of modal representations and the utilization of multimodal information are relatively simplistic and crude. The paper of MECPE-2steps [31] also mentions this issue and similar observations have been noted elsewhere These systems are still

inadequate for deep understanding and reasoning in emotion consequence forecasting in conversations, leaving substantial room for improvement. Furthermore, employing a clue-driven hybrid method may enhance the comprehension of multimodal information and the relationships between utterances.

*6.3.3* ***Effectiveness of the Clue-Driven Hybrid Methods.*** We conduct the clue-driven hybrid experiments to explore the effectiveness of the clues from the MLLMs. We can see that in the majority of cases, the clue-driven hybrid methods have shown a significant improvement in the F1 score. Especially in the ECFCON-BERT, the clue-driven hybrid method has further improved by 0.45%, 2.92%, and 2.49%, respectively, in the CF, ECPF, and ECPF-C tasks.

The clue-driven hybrid method greatly compensates for the shortcomings of the traditional methods in handling multimodal information.

• Firstly, with the help of the MLLMs, it is relatively easy to extract the required clues from videos, such as facial expressions, action scenes, etc. These clues are not involved in traditional feature construction, and using specialized models for facial or action extraction is overly complex and computationally expensive. Despite this, the performance may not be as well as MLLMs.

• Secondly, there is a lack of overall understanding of video conversations. Traditional methods involve segmenting the video into fragments and extracting features from different modality extractors, which introduces a lot of modality gaps and biases. Inputting the entire video into the MLLMs may greatly alleviate this issue. The clues extracted in this way are more accurate and meaningful.

• Finally, inputting the content of the entire dialogue into the MLLMs and then asking *emotion*, *impact*, and *why* questions may excellently mine the inherent conversational logic and common sense knowledge in MLLMs.

In addition, some cases have shown a decline in performance after adding the clues, which may be due to the following reasons:

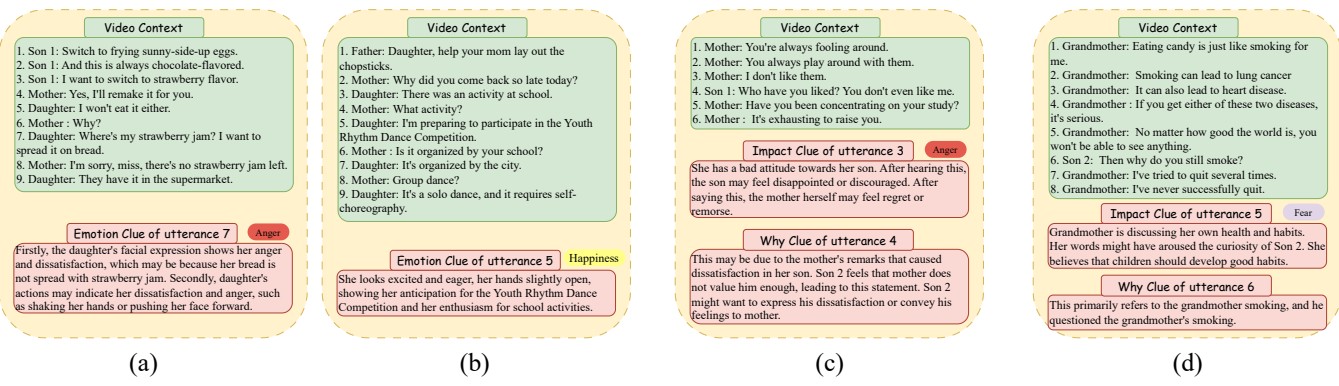

Figure 5: Four cases of the clues from the MLLMs.

• Firstly, for STGraph, it is difficult to handle the *impact*, and *why* clues to construct the before and after relationship of the conversation. The clues are mixed together, and obviously, this chaos leads to a decline in performance.

• Secondly, in the remaining cases, only in the ECPF task, ECFCON-LSTM and ECFCON-RoBERTa perform abnormally. This fluctuation comes from the fact that the subtask of emotion recognition is relatively simple, requiring only the recognition of whether there is an indicative emotion. This subtask can also be competently handled by traditional tasks to some extent.

• Finally, the decline in the clue-driven hybrid method may be partly due to some bias in the clues. The parameters of our MLLMs are only 7B, and if a model with better performance is used, such as the 13B model, the quality of the clues may be further improved and perhaps this situation will not happen.

### 6.4 Few-shot Learning

To investigate the inference capabilities of MLLMs and the effectiveness of the clue-driven hybrid methods, we conduct numerous few-shot learning experiments on the ECFCON dataset. Figure 4 shows the performance of these methods in the CF, ECPF, and ECPF-C tasks respectively. Here, the data size denotes the number of dialogues used for fine-tuning the MLLMs.

From the figure, it can be seen that the generalized prompting methods directly surpass other methods in the zero-shot setting, but in the few-shot setting, the performance is not as good as expected. One reason is that a small number of samples disrupts the knowledge structure of the MLLMs and brings much bias to the model, leading to a rapid decline in performance. Another reason is that for consequence forecasting, the MLLMs may be limited in understanding the definition of consequences. Fine-tuning with a small size of samples may lead the model into a deep abyss of incorrect understanding. This decline in performance leads to the outcome that a clue-driven strategy is chosen to alleviate this situation. Clues are inherent in MLLMs and can be extracted much easier to obtain, which is more accurate and meaningful.

Furthermore, we also compare the performance of traditional and clue-driven hybrid methods in few-shot learning. The light-colored lines denote the traditional methods, while the dark-colored lines denote the clue-driven hybrid methods. It can be seen that the clue-driven hybrid methods outperform traditional methods in the

majority of cases. For instance, in Figure (b), the line of the ECFCON-BERT (clues) and MECPE-2steps (clues) are overall above the line of the ECFCON-BERT and MECPE-2steps. This demonstrates that clues are effective across different sizes of training sets and can greatly improve the performance of the traditional methods.

### 6.5 Case Study

We present four cases of the clues from the MLLMs in Figure 5 (a-d).

From (a) and (b), it can be seen that the emotional clues are extracted from the textual content, the facial expressions, and the action scenes of the video. These clues can provide a wealth of meaningful information for emotion recognition. For instance, in (a), the facial expressions of the daughter are very angry, and her actions, such as shaking her hands and pushing her face forward, indicate her dissatisfaction. If there are no clues, inferring emotions solely based on the content of the dialogue is challenging.

From (c) and (d), it can be seen that the *impact* and *why* clues are extracted to describe the relationship between the emotional utterance and the subsequent utterance. These clues can assist in understanding the logic between the proceedings and the following parts of the conversation. For instance, in (d), the *impact* clue indicates that the grandmother's words may arouse the curiosity of Son 2, and the *why* clue indicates that Son 2's words are because of the grandmother's smoking. Then, these two clues can be united to help to forecast the consequences more accurately.

## 7 CONCLUSION

In this paper, we have introduced the task of ECFCON, and annotated a new dialogue-level video dataset, containing 2,780 video dialogues and 12391 emotional utterances, of which 8,810 have consequences. Then, we have benckmarked this task by traditional methods, generalized LLMs prompting methods, and clue-driven hybrid methods, where the last one performed the best.

In our future work, we plan to further annotate the concrete and specific content of consequences in utterances, which can be seen as a fine-grained consequence forecasting task. Besides, we will jointly annotate the cause, emotion, and consequences, thereby constructing the complete chains from cause to emotion to consequences.

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
