# OpenReview forum: "ECFCON: Emotion Consequence Forecasting in Conversations"
_acmmm.org/ACMMM/2024/Conference — MM2024 Poster_

### Official Review · Reviewer_NfgP · 2024-05-10

**Rating:** 5
**Confidence:** 3

**Summary:**

The paper proposes a new task called Emotion Consequence Forecasting in CONversations (ECFCON), which involves forecasting the consequences of emotions in dialogues. The ECFCON contains three sub-tasks:  Consequence Forecasting (CF), Emotion Consequence Pair Forecasting (ECPF), and Emotion Consequence Pair Forecasting with Categories (ECPF-C). The video dialogues are collected from a Chinese situation comedy, which includes 2,780 video dialogues with a total of 39,950 utterances. To provide a comprehensive benchmark, this paper proposes three types of baselines: traditional methods, generalized LLMs prompting, and clue-driven hybrid methods. Numerous experimental results demonstrate the potential value of the task.

**Strengths:**

This paper introduces a new dimension to the study of emotions in conversations by focusing not just on the recognition of emotions or their causes, but on forecasting their consequences on subsequent interactions. This is a novel approach that expands the scope of emotional AI research. The ECFCON dataset includes a substantial number of video dialogues with emotional utterances and their consequences, which can serve as a benchmark for future research in the field. The clear presentation of the task, dataset, and experimental results is a strength as it allows for easy understanding.

**Limitations:**

The main contribution of this paper is to propose a new task called Emotion Consequence Forecasting in conversations. However, the emotion forecasting and the emotion cause recognition are reversed tasks each other. Firstly, what benefits do forecasting the consequences of emotions in conversations provide? Secondly, why can't invert the emotion cause dataset into the emotion forecasting dataset? Furthermore, since the newly proposed dataset is a multimodal dataset, the traditional methods lack the module to explicitly fuse multimodal information.
Detailed comments:
1. The line in Figure 4 is not clear and needs a thicker line.
2. The clue-driven hybrid method relies too heavily on text modals.
3. Possible formatting error in reference 1.

**Suitability:**

3

---

### Official Review · Reviewer_tuLe · 2024-05-19

**Rating:** 4
**Confidence:** 3

**Summary:**

This paper focuses on multimodal dialogue emotion analysis, and proposes a new task Emotion Consequence Forecasting in conversation. For the task, this paper builds a Chinese video dialogue-level dataset, and constructs a framework ECFCON, which combines multimodal video-based large language models with traditional models using generalized prompting methods and clue-driven hybrid methods. This paper setups experiments for proving the performance of ECFCON, and analyzes the experimental results.

**Strengths:**

1. Emotion Consequence Forecasting proposed by this paper is a novel task and may be significant for emotion dialogues.
2. This paper builds a new video dialogue dataset for the Emotion Consequence Forecasting task, which is conducive to further research in the field of emotional dialogue.
3. The framework proposed in this paper combines the capabilities of large models while exploiting the advantages of traditional models.

**Limitations:**

1. Emotional cause analysis can play a role in human-computer empathy and positive emotion guidance in dialogue. As a new task, the role of Emotion Consequence Forecasting in conversation does not seem to be very clear, or is not clearly articulated in this paper.
2. In experiments, P, R and F1 are all commonly used evaluation metrics, but the three tasks proposed in this paper are all prediction probabilities. In this case, how do P, R, and F1 be calculated? (For example, is 70% predicted correctly or incorrectly?)
3. According to Table 3, comparing with Effectiveness of the Clue-Driven Hybrid Methods, Modality Effectiveness does not seem obvious.
4. In dialogues, some works such as empathy dialogue and emotional support dialogue should be more related with the task of Emotion Consequence Forecasting than emotion recognition, the works should be mentioned in Related Work.

**Suitability:**

3

---

### Official Review · Reviewer_QcBx · 2024-05-19

**Rating:** 4
**Confidence:** 3

**Summary:**

This paper proposed a new task in emotion extraction in conversation, the idea of this work is good, and also has sufficient evaluation about this task, while some limitations should be concerned

**Strengths:**

The idea of emotion consequence forecasting is intriguing, and this paper also provides datasets for this task.

**Limitations:**

(1) However, it's not entirely clear how the 'clues' method differs from tasks like emotional cause extraction, as discussed in 'ECPEC: Emotion-Cause Pair Extraction in Conversations.'
(2)In the framework, the process for selecting feature extraction models isn't clearly explained. Why were HuBERT, LSTM, and CLIP chosen? Are the features extracted by these models superior to others? Is there a specific rationale behind this selection? Actually, this makes the work incremental.
(3)The paper lacks a definition for variable 'h' in equations 6, 9, and 20.
(4)The text in figure 2 and the labels in figure 4 are too small, making them difficult to read.
(5)The captions accompanying the figures and tables lack necessary explanations. For instance, in figure 2, it's unclear what the proposed framework is, and the method is presented confusingly. Also, there's a typo: 'traditional' should be spelled correctly.

**Suitability:**

3

---

### Official Review · Reviewer_Foa8 · 2024-05-22

**Rating:** 3
**Confidence:** 3

**Summary:**

This paper introduces a new task of emotion consequence forcasting (ECF). A multi-modal (text/audio/video) dataset is constructed for this task, and several approaches have been developed to solve this task.

**Strengths:**

1. The task of ECF is new.
2. An ECFCON dataset is built for the study of this task.

**Limitations:**

1. The correctness of the definition of the task should be further clarified. In my opinion, it is difficult to determine that one utterance is the consequence of the emotions of its previous utterances, since an utterance contains rich information, such as content, emotion, body movement, etc and its causes should be more complex than simple the emotions of previous utterances.
2. In Section 3, I can't understand why the first consequence type is objective and the second one is subjective. In my opinion, they are both subjective since they are annotated by humans.
3. In the last paragraph of Section 3, how many emotion categories are used for Task3?
4. Please clarify the details of the results in Table 1.

**Suitability:**

3

---

### Meta-Review · Area_Chair_964k · 2024-07-02

**Recommendation:** Accept (Poster)
**Confidence:** 4

**Metareview:**

This paper proposes a new task called Emotion Consequence Forecasting in Conversations (ECFCON), aiming to predict the emotional consequences of a conversation. A multimodal dataset, comprising text, audio, and video, 12,391 emotion-annotated utterances, from which 8,810 are annotated with consequences, is proposed.

The paper got somewhat mixed reviews, but tending towards acceptance, with one Weak Accept, two Borderline Accepts, and one Weak Reject.

As strengths, reviewers recognize the novelty and relevance of the proposed task for the multimedia community and value the introduction of a multimodal dataset to conduct benchmarking.

However, some limitations are evidenced. Namely:
- The concept of the task could be better articulated (Foa8, tuLe, NfgP)
- Some parts of the approach are not well explained (QcBx)
- Some readability issues were pinpointed by reviewers Foa8 and QcBx.

While the authors attempted to address these concerns in their rebuttal, all reviewer scores remained unchanged.

Given the relevance of the proposed task and the potential impact of the proposed dataset, despite the paper's limitations, the paper has its merits. Therefore, I suggest this work to be accepted as Poster, provided that authors accommodate the discussed clarifications mentioned in the rebuttal, and proofread the paper to fix the typos/grammar issues identified by the reviewers.